# CAR-T Cell Therapy for Acute Myeloid Leukemia: Where Do We Stand Now?

**DOI:** 10.3390/curroncol32060322

**Published:** 2025-05-30

**Authors:** Pilar Lloret-Madrid, Pedro Chorão, Manuel Guerreiro, Pau Montesinos

**Affiliations:** 1Hematology Department, Hospital Universitari i Politècnic La Fe, 46026 Valencia, Spain; pilar_lloret@iislafe.es (P.L.-M.); pedro_chorao@iislafe.es (P.C.); manuel_direito@iislafe.es (M.G.); 2Instituto de Investigación Sanitaria La Fe (IISLAFE), 46026 Valencia, Spain; 3Department of Medicine, University of Valencia, 46010 Valencia, Spain

**Keywords:** acute myeloid leukemia, relapsed or refractory AML, CAR-T cell therapy

## Abstract

**Background**: Patients with refractory and relapsed acute myeloid leukemia (R/R AML) face a dismal prognosis. CAR-T therapy has emerged as a potential treatment option. This study assesses the available clinical evidence on CAR-T in R/R AML, focusing on safety and efficacy outcomes. **Methods**: We included studies on CAR-T therapy for R/R AML published from June 2014 to January 2025. Data on patient and disease characteristics, CAR-T constructs, response rates, post-CAR-T allogeneic HSCT (allo-HSCT), and safety outcomes were analyzed. **Results**: Twenty-five CAR-T clinical trials involving 296 patients were identified. The most frequently targeted antigens were CD33, CD123, and CLL-1, while CD7, CD19, NKG2D, and CD38 were also explored. Responses were heterogeneous and often short-lived when not consolidated with allo-HSCT. Cytokine release syndrome and neurotoxicity were generally low grade and manageable. Prolonged and severe myelosuppression was a frequent limiting toxicity, often requiring allo-HSCT to restore hematopoiesis. Disease progression was the leading cause of death, followed by infections. **Conclusions**: CAR-T cell therapy may represent a feasible therapeutic strategy, particularly as bridging to allo-HSCT to mitigate myelotoxicity and improve long-term outcomes. Nevertheless, it remains in the early stages of development and faces significant efficacy and safety challenges that must be addressed in future trials to enable the expansion of this promising therapeutic approach for a population with high unmet medical needs.

## 1. Introduction

Acute myeloid leukemia (AML) is an aggressive blood cancer characterized by the uncontrolled proliferation of immature myeloid lineage cells, leading to impaired hematopoiesis and bone marrow dysfunction [1]. AML is the most common acute leukemia in adults and accounts for approximately 1% of all cancers in the United States [2]. Although intensive chemotherapy induces complete remission (CR) in approximately 60–80% of adult patients, more than half will eventually relapse [3,4,5]. Therefore, refractory or relapsed (R/R) AML is a common scenario, with a 3-year estimated survival of only 10% [6]. Allogeneic hematopoietic stem cell transplantation (allo-HSCT) provides the best chance for long-term survival in R/R AML patients. However, achieving a CR prior to transplantation is often a prerequisite at most transplant centers, and a substantial proportion of patients are unable to attain remission with currently available therapies. In addition, around 40% of patients who undergo HSCT will subsequently relapse [5,7]. Patients relapsing after HSCT face a particularly poor prognosis, with a long-term survival rates below 20% [8,9]. These data underscore the urgent need for novel treatment approaches for R/R AML.

Chimeric antigen receptor T (CAR-T) cell therapy has emerged as a potential treatment for R/R B-cell hematologic malignancies, including acute lymphoblastic leukemia (ALL), B-cell lymphomas, and multiple myeloma [10,11,12,13]. Although CAR-T therapies for AML have been actively explored in recent years by targeting several antigens, they have yet to achieve widespread clinical success. Several challenges have been identified from early-phase clinical and preclinical studies, including antigenic heterogeneity, on-target off-tumor toxicity, and the inability of CAR-T cells to expand and persist due to the immunosuppressive microenvironment of AML [14,15,16,17]. As a result, no CAR-T product has yet demonstrated sufficient safety and efficacy to warrant regulatory approval in AML.

This systematic review aims to summarize the available evidence on CAR-T therapies in AML, analyzing the various investigational cellular products currently being tested in human clinical studies, along with their safety profile, efficacy outcomes, and the potential role of CAR-T as a bridge to allo-HSCT.

## 2. Materials and Methods

### 2.1. Search Strategy

This systematic review was conducted in accordance with the PRISMA 2020 guidelines and was registered in PROSPERO (registration number: CRD420251024766) [18,19]. The PRISMA 2020 checklist is provided as Appendix A. Initially, a comprehensive search was conducted using PubMed’s advanced search features, employing specific keywords and Boolean operators to cover publications from the past decade as follows: ((acute myeloid leukemia) AND ((CAR-T) OR (chimeric antigen receptor T-cell))) AND ((“6 January 2014”[Date—Publication]: “1 January 2025”[Date—Publication])). Subsequently, searches were carried out on clinicaltrials.gov with “acute myeloid leukemia” as the condition and “chimeric antigen receptor T-cell” or “CAR-T” as the intervention” to identify both completed and ongoing human clinical trials with published results. Finally, abstracts from congresses held by the following five major medical societies over the last seven years (2017–2024) were reviewed: the European Hematology Association (EHA), American Society of Hematology (ASH), European Society for Blood and Marrow Transplantation (EBMT), American Society for Transplantation and Cellular Therapy (ASTCT), and American Society of Clinical Oncology (ASCO).

### 2.2. Inclusion and Exclusion Criteria

The inclusion criteria used were as follows: (a) original research articles, abstracts or other published trial/study results involving human subjects; (b) results published in peer-reviewed journals or congresses with peer-reviewed abstract acceptance; and (c) texts available in English. The exclusion criteria were as follows: (a) preclinical studies either in vitro or using animal models; (b) studies focused on CAR-NK cell therapy; (c) review articles, editorials, or comments; and (d) case reports or abstracts that were later incorporated into larger studies (when multiple reports from the same study were available, only the most complete and updated version was considered).

### 2.3. Data Extraction and Synthesis

Data extraction was performed independently by two reviewers (P.L. and P.C.). PubMed search results were reviewed using the Rayyan webapp [20]. Duplicates were automatically detected and were user-checked before definite exclusion. All other results from different sources were manually reviewed. Discrepancies or disagreements were resolved through discussion and consensus between all authors. Prior to information extraction and tabulation, all authors reviewed and agreed upon the final list of articles/abstracts.

Figure 1 shows the PRISMA 2020 flow diagram for systematic reviews, illustrating the workflow carried out.

### 2.4. Variables Selected and Outcome Measurement

We collected data from clinical trials, including publication year, study phase, and country. We also reviewed baseline demographic characteristics of patients, as well as clinical and biological features of AML, such as genetics, disease status, and the number of prior lines of treatment. Additionally, we assessed characteristics of the CAR-T product, including source (autologous/allogeneic), target antigen, apheresis, manufacturing time, transduction mechanism and co-stimulatory domain, lymphodepleting regimens, prior HSCT and subsequent allo-HSCT post-CAR-T, whenever available.

The primary objective variables were the overall response rate (ORR) [including CR, complete response with incomplete hematologic recovery (CRi), and morphologic leukemia-free state (MLFS)], which was defined exclusively based on morphological criteria regardless of minimal residual disease status; the rate of allo-HSCT post-CAR-T; and the safety outcomes related to CAR-T therapy, including treatment-related mortality, cytokine release syndrome (CRS), immune effector cell-associated neurotoxicity syndrome (ICANS), related cytopenias, infections, and graft-versus-host disease (GvHD).

Due to the limited number of studies on this subject, a meta-analysis was not feasible. In addition, we performed a descriptive assessment of study quality and potential risk of bias across all included studies. A summary of this analysis is provided in Appendix A. Missing data were documented as such, and ambiguous information was reviewed by the authors to decide whether to include or exclude it from the review.

## 3. Results

### 3.1. Targets and Constructs Under Development

Table 1 outlines the main characteristics and roles of the proposed target antigens for CAR-T therapy in AML. CD33 and CD123 antigens are among the most frequently studied antigens, but their expression on hematopoietic stem cells (HSCs) represents a potential risk of myelotoxicity [21,22,23,24]. To address this limitation, other antigens with minimal or no expression on HSCs are being targeted in preclinical and/or clinical phases, including CLL-1, CD38, CD7, CD84, CD93, CD70, IL-1RAP, and folate receptor β [24,25,26,27,28,29,30,31,32,33,34,35,36,37,38,39,40,41].

### 3.2. Clinical-Phase CAR-T Constructs

A total of 25 clinical-phase studies with available results were included: 15 from the database search and 10 from congress abstract books. Table 2 summarizes the key characteristics of the CAR-T constructs, while Table 3 outlines the disease and patient characteristics, efficacy outcomes, and safety results of CAR-T cell therapies based on these studies.

Overall, 25 different CAR-T constructs have been evaluated, most originating from China and the United States (Table 2). CD33, CD123, and CLL-1 were the most targeted antigens due to their wide expression on leukemic stem cells (LSCs) and leukemic blasts [63,64,65,66,67,68,69,70,71,72,73,74,75,76,77,78]. However, clinical studies have also explored other myeloid antigens with lower expression on HSCs, such as CD38 and NKG2D-L, as well as aberrantly expressed lymphoid antigens like CD19 and CD7 [27,39,42,60,61,62,79,80,81]. In this regard, it is important to note that CD19, a key marker in B-cell differentiation, is aberrantly expressed in leukemic blasts in about 8% of AML cases, but its expression is notably higher in AML with t(8;21), where it is found in 50–93% [62,82]. In terms of cell sources, autologous CAR-T therapies remain the predominant approach, although there is increasing use of allogeneic CAR-Ts. Regarding manufacturing times, reported durations ranged from 8 to 14 days. The most common co-stimulatory domain used was 4-1BB-CD3ζ, with some clinical trials incorporating novel modifications to the CAR-T design, such as genetic knock-out of PD-1, MICA/B, or autologous T-cell receptor (TCR), aimed at improving cell expansion and reducing toxicity, as mentioned below [60,67,69,70,76,78,79,81].

#### 3.2.1. Patients and Disease Characteristics

The available data on the characteristics of patients treated in the included studies show considerable heterogeneity (Table 3). Patient age ranged from pediatrics to older adults (median age from 12 to 70 years). Regarding the number of previous therapies, in all cases patients were heavy treated, with three to five previous lines in the majority of studies, including allo-HSCT. The presence of extramedullary disease (EM) was not available in most studies, but ranged from 14% to 43%. Regarding the use of CAR-T therapy in the setting of EM, it is worth highlighting the study by Zhao et al. [75], which included a total of 20 patients with EM (43%), with the following distribution: CNS (*n* = 12), liver and spleen (*n* = 3), skin (*n* = 2), ovary (*n* = 1), kidney (*n* = 1), and breast (*n* = 1).

#### 3.2.2. Bridging Therapy and Lymphodepletion

Most studies do not provide details on whether patients were bridged before CAR-T, and the bridging therapy used, leaving this information scarce and highly variable. One study mentions that bridge chemotherapy was not required due to rapid CAR-T manufacturing, as it was produced locally [62]. Another study by Ma et al. [76] indicates that bridging therapy was used to reduce tumor burden prior to CAR-T CLL-1 infusion in the two enrolled patients. The first patient underwent D-CLAG (decitabine, cytarabine, cladribine, and granulocyte colony-stimulating factor), while the second received CLAG alone.

Regarding conditioning prior to CAR-T therapy, the majority of clinical trials used a regimen based on cyclophosphamide (Cy; 300–500 mg/m^2^) and fludarabine (Flu; 25–30 mg/m^2^) on days −5 to −3 (Table 3). Within this regimen, some trials included novel modifications, such as the addition of alemtuzumab, etoposide or decitabine [28,60,61,70]. In contrast, five clinical studies did not use lymphodepletion [63,65,79,80].

#### 3.2.3. Toxicity Profile

Regarding immune-mediated inflammatory adverse events, the incidence of CRS ranged from 66.7% to 100%, with most cases being grades 1–2. Neurotoxicity was less frequent, occurring in 0% to 50% of patients. Similar to CRS, cases of severe ICANS (grade 3–4) were infrequent (Table 3). Both toxicities were reversible and manageable with conventional treatment (tocilizumab and corticosteroids). Only one case of CRS with a fatal outcome was reported, with no deaths attributable to neurotoxicity [70]. Furthermore, the presence of EM may not significantly affect CAR-T-related toxicities, as the study by Zhao et al. [75] found comparable incidence rates of complications between the cohorts with and without EM.

Concerning GvHD, although most studies did not provide specific data, those that included allogeneic CAR-T cells show mixed results. Some studies reported no GvHD cases [60,77], while others observed adverse events, such as one case of mild acute cutaneous GvHD, two cases of grade 1–2 GvHD, and one fatal case of grade 4 hepatic and gastrointestinal GvHD, all of which occurred either during or after allo-HSCT [42,67,71,73].

On the other hand, serious infectious complications were a significant concern, ranking as the second leading cause of mortality after disease progression, with sepsis being the most common underlying factor [60,62,73,75,80]. In the study conducted by Zhao et al. [75], 40 out of the 47 patients (85%) who received CAR-T infusion developed infections, including 37 with bacterial infections, 10 with fungal infections, and nine with viral infections. Among the studies with available data, the causes of death were distributed in the following occurrence: disease progression, infections, GvHD, and other causes, including cerebral hemorrhage and Epstein-Barr virus-associated B-cell lymphoma [39,42,60,62,63,64,73,75,80].

Data on myelotoxicity were limited across studies. When reported, the incidence of post-CAR-T cytopenias varied from 12.5% to 100%, varying by target antigen. Overall, a high frequency of myelotoxicity was observed, with some studies reporting grade 4 pancytopenia in all patients [28,39,60,67,75,76,77,78]. In one study evaluating CD123-directed CAR T cells, all cytopenias were transient, suggesting potential reversibility [72]. Many patients underwent allo-HSCT within a few weeks after CAR-T infusion, making it challenging to accurately assess the duration of cytopenias and hematopoietic recovery, as these may be affected by the subsequent conditioning regimen and engraftment following allo-HSCT. However, in the study by Wermke et al. [69], using a switchable CD123 CAR-T, no long-term treatment-related myelosuppression was observed, and none of the patients required allo-HSCT for hematologic recovery.

#### 3.2.4. Efficacy Results

The ORR (CR/CRi/MLFS), varied across studies, ranging from 0% to 100% (Table 3). Given the heterogeneity of patient cohorts and sample sizes, direct comparisons between CAR-T constructs are not feasible. To offer a clearer view, key preliminary efficacy outcomes from the most representative studies are summarized below, stratified by CAR-T antigen target.

CD7: In the study by Hu Y et al. [61], in which anti-CD7 CAR-T cells were evaluated in CD7-positive R/R AML, a CR/CRi rate of 66.7% was achieved. (6/9). Among responders, three proceeded to allo-HSCT, and five remained in remission at a median follow-up of 5.4 months. A key challenge with CD7-targeted CAR-T therapy is the expression of CD7 on normal lymphocytes, which can lead to CAR-T cell fratricide. To address this, Gomes-Silva et al. [83] showed that using CD7-targeted CAR-T cells derived from CD7 gene knockout T cells prevents fratricide and enhances CAR-T expansion and persistence.

CD19: In a phase 2 study, CAR CD19 T-cell therapy was administered to six patients with CD19-positive R/R AML, most of whom had t(8;21) translocation [62]. Four of the six patients (66.7%) achieved CR/CRi, while two had disease progression. Among responders, two subsequently underwent allo-HSCT. The median duration of remission in patients achieving CR/CRi was 8.5 months (range, 3–14).

CD33: Sallman et al. [65] conducted a phase I/Ib trial, treating 20 patients with AML and four patients with other myeloid malignancies. The ORR was 10% (2/20), with one patient remaining in remission for 18 months post-allo-HSCT. In another recent study, five of the 12 patients infused achieved CR/CRi (41.7%), and four of these responders were bridged to allo-HSCT [68].

CD38: In the study conducted by Cui Q et al. [39], six patients with R/R AML, all of whom had relapsed after allo-HSCT, were enrolled. CR/CRi was achieved in 66.7% (4/6). However, the median remission duration was 191 days (range 117–261 days), with a 6-month relapse incidence of 50%.

CD123: In the clinical trial conducted by Wermke et al. [69], 19 patients with R/R AML were treated, yielding an ORR of 53% (8/15 evaluable patients). Another phase I/Ib trial, reported an ORR of 12.5% (2/16 patients), with one achieving MRD-negative CR and persisting remission for 8 months post-treatment [70]. These studies, as shown in Table 3, demonstrate significant variability in responses, ranging from 12.5% to 50% for this antigen.

CLL-1: Zhao et al. [75] reported an ORR of 74.5% in 35 out of 47 patients, with 65% in those with EM and 81.5% in those without EM (*p* = 0.31). Although the overall survival (OS), progression-free survival, and duration of remission seemed shorter for patients with EM compared to those without, this difference was not statistically significant. Studies involving CLL-1-targeted CAR-T therapy show ORR ranging from 70% to 100%, as presented in Table 3.

CLL-1/CD33: In the study by Fang Liu et al. [78] using a dual-target CAR-T, the ORR was 78% (7/9 patients), all of whom achieved MRD-negative CR. Subsequently, six of the responders were bridged to allo-HSCT.

NKG2D-L: The study by Sallman et al. [79] represents the largest trial targeting NKG2D-L to date. This multicenter, open-label, dose-escalation phase study included 25 patients. Among the 12 evaluable patients with R/R AML (*n* = 11) or myelodysplastic syndrome (MDS) (*n* = 1), three achieved CR/CRi, resulting in an ORR of 25%. Of the responders, two patients with R/R AML proceeded to allo-HSCT and remained in remission at 5 and 61 months. In contrast, other clinical studies reported no remissions [80,81].

## 4. Discussion

Selecting an appropriate target for CAR-T therapy in AML presents a significant challenge due to the disease’s broad heterogeneity and the variable expression of antigens on malignant cells. Another key limitation is the expression of target antigens on both leukemic and non-leukemic hematopoietic cells, which raises concerns about potential myeloablation and the need for salvage with allo-HSCT. Several clinical trials have explored CAR-T therapies targeting multiple antigens in AML, and preliminary results suggest that targeting broadly expressed antigens (such as CD123, CD38, CLL-1, NKG2D-L or CD33) or less common ones (such as CD7 and CD19) could lead to clinical responses in this difficult-to-treat population. However, the identification of optimal antigen combinations that ensure efficacy while minimizing hematopoietic toxicity, as well as predictive biomarkers of response, remain unresolved areas with scarce clinical and preclinical evidence.

We acknowledge several limitations in our review, including the absence of randomized controlled trials, small sample sizes, high heterogeneity, and short follow-up periods, which limit the ability to summarize the evidence and make direct comparisons between studies. Additionally, a significant part of the data comes from conference abstracts and case reports, which provide limited information and may affect the overall strength of the evidence available to date. Furthermore, it is important to highlight that all the presented studies lack controlled arms, which further limits the interpretation of efficacy results, especially when compared to historical salvage therapies. This limitation underscores the need for future research with more robust study designs, such as studies including controlled arms with current salvage standard therapies, to better assess the true clinical benefit of CAR-T therapies in AML.

In the following sections, we discuss the main challenges of CAR-T therapy in AML, both during the production phase and at different clinical stages, along with potential solutions to address them, as illustrated in Figure 2.

### 4.1. Challenges Related to CAR-T Cell Construct Design and Application in AML

A major challenge in applying CAR-T cell therapy is antigen escape [84]. Patients receiving multiple cycles of chemotherapy may experience a reduction in the expression of target antigens, which can decrease CAR-T effectiveness [85,86,87]. To mitigate this problem, multiple strategies were developed. One promising approach involves dual-targeting CAR-T cells, which are engineered to recognize two distinct antigens simultaneously, thereby reducing the risk of tumor evasion [87,88,89]. For example, Fang Liu et al. [78] used a dual CAR-T targeting CLL-1 and CD33, reporting promising preliminary efficacy outcomes (ORR 78%). Other approaches under development include the use of drugs like all-trans retinoic acid and hypomethylating agents, which upregulate antigen expression in preclinical studies [90,91,92], potentially improving CAR-T target recognition.

Moreover, to address CAR-T cells’ limited persistence and functionality, next-generation constructs have been developed to optimize intracellular signaling and enhance antitumor efficacy. Third-generation CAR-T cells include two co-stimulatory domains, which enhance activation and promote tumor elimination. Fourth-generation CAR-T cells, also known as “armored CARs”, contain additional transcription factors that boost the production of inflammatory cytokines, potentially boosting the immune response. Finally, fifth-generation CAR-T cells integrate an IL-2Rβ domain, which activates the JAK/STAT pathway, improving T-cell proliferation and persistence [89]. Some of these have already been tested in human clinical trials, such as PRGN-3006 UltraCAR-T and CYAD-02 CAR-T [65,81]. PRGN-3006 UltraCAR-T combines CD33-targeted CAR-T cells with membrane-bound IL-15 to enhance cell persistence, while CYAD-02 CAR-T uses a non-gene-editing approach to silence the expression of MICA and MICB, stress-induced ligands that are overexpressed on CAR-T cells and can trigger fratricide. Additionally, CAR-NK cells have emerged as a promising alternative to CAR-T cells for AML treatment due to their innate cytotoxicity and secretion of high levels of interleukins, particularly interferon gamma (IFN-γ) [93]. Unlike CAR-T cells, CAR-NK cells can be used allogeneically without inducing GvHD, as they target AML cells that downregulate human leukocyte antigen I (HLA-I) expression. Additionally, CAR-NK therapy is associated with lower rates of CRS and ICANS compared to CAR-T [94]. However, challenges remain, including a short half-life and manufacturing complexity, which may require multiple infusions [95]. Several clinical trials, not covered in this review, are investigating CAR-NK therapy for AML, with two completed studies reporting variable outcomes: Huang et al. [96] achieved a 20% complete remission rate with low-grade CRS, while Tang et al. [97] noted low toxicity but observed disease progression in all patients. Further research is needed to optimize the promising potential of CAR-NK therapy for AML treatment.

The tumor microenvironment in AML also poses a challenge to CAR-T cell efficacy, primarily due to the accumulation of myeloid-derived suppressor cells and regulatory T lymphocytes (Tregs), which hinder T cell activation [98,99]. Similar challenges are also observed in other myeloid neoplasms, such as MDS, where antigen heterogeneity, immunosuppressive bone marrow microenvironment, and CAR T-cell exhaustion have been highlighted in a recent review [100]. To overcome this, strategies, such as combining CAR-T therapy with immune checkpoint blockade (e.g., PD-L1 or CTLA-4 inhibitors), are being explored to improve CAR-T persistence and efficacy [88,101]. Another promising approach involves genetically modifying CAR-T cells to eliminate immune checkpoint molecules. For example, Ma et al. [76] targeted CLL-1 with PD-1 silencing in CAR-T cells for R/R AML. By silencing PD-1 using small interfering RNA (shRNA), which is a short RNA sequence that inhibits the expression of specific genes, CAR-T cells were able to bypass the PD-1/PD-L1 suppression, enhancing their anti-leukemic activity and preventing exhaustion for a more durable response.

### 4.2. Manufacturing Failures and Delays

AML primarily affects older patients who have often undergone multiple prior treatment regimens, which can impair T cell function and CAR-T manufacturing [88,102]. Furthermore, the presence of blasts in peripheral blood has been shown to impact T cell quality, leading to delays and failures in CAR-T production. In the study by Sallman et al. [79], seven out of 25 patients experienced manufacturing failures. Other studies reported a waiting period of 36 to 94 days between apheresis and CAR-T infusion, further underscoring the significant challenges in manufacturing [63]. However, the PRGN-3006 UltraCAR-T clinical trial employed an advanced, non-viral gene delivery system coupled with a rapid, decentralized manufacturing process. This enabled overnight CAR-T cell production at medical centers, using patients’ autologous T cells [65]. The cells were administered just one day after gene transfer, without the need for ex vivo activation or expansion. This approach not only reduces waiting times but also simplifies logistics, making it particularly suited for urgent AML treatments. Another promising approach involves using allogeneic CAR-T cells from healthy donors or prior transplant donors. This method eliminates the need for modifying the patient’s own T cells, potentially simplifying the manufacturing process and expanding CAR-T therapy as an “off-the-shelf” treatment. However, this approach has significant limitations compared to autologous CAR-T therapies. In particular, genetic modifications are required to prevent GvHD and allogeneic rejection. Examples of such modifications include knockout of the endogenous TCR to reduce the risk of GvHD, and deletion or modification of HLA molecules to evade host immune recognition. Some of these approaches have reached clinical trials and have been investigated in AML, as will be discussed below [70].

### 4.3. Bridging Therapy

AML rapid progression and high risk of life-threatening infections contrasts with the current 2 to 6 weeks manufacturing interval, during which patients may experience deterioration in performance status and, in some cases, death. This highlights the urgent need for effective bridging strategies to control disease until CAR-T therapy. This was demonstrated in a study where three out of 23 patients (13%) did not receive the infusion due to rapid disease progression [66]. Although most trials do not specifically detail bridging chemotherapy, data from Ma et al. [76] suggest potential benefits in reducing tumor burden. For example, a review of 62 adults with B-ALL showed that bridging chemotherapy before CAR-T led to a median OS of 16.3 months, compared to 4.3 months for those who did not receive it (*p* = 0.04). However, high-intensity chemotherapy did not improve OS and was associated with increased infections and toxicity [103]. We can speculate that debulking AML before CAR-T infusion could be beneficial in reducing immune-mediated toxicities and/or mitigating the risk of resistance or relapse. Unfortunately, unlike ALL, R/R AML is less responsive to low-dose cytotoxic regimens, which limits the potential of this approach.

### 4.4. Lymphodepletion

Lymphodepletion plays a crucial role in facilitating CAR-T cell expansion and engraftment by reducing endogenous T cells and immunosuppressive cells, such as Tregs [88,104,105]. While some trials have explored omitting lymphodepletion, preliminary evidence in AML supports its use. For example, in a trial comparing two cohorts with and without lymphodepletion (using Cy + Flu), patients who received lymphodepletion exhibited higher CAR-T expansion and CR rates [65]. Although Cy + Flu and its variations are most commonly used in the clinical trials for hematological neoplasms, the choice of regimen should be tailored to the patients’ conditions and comorbidities, taking potential toxicities into account.

### 4.5. CAR-T-Related Toxicity

As with other CAR-T therapies, the most common related adverse events associated with CAR-T therapy in AML are CRS and ICANS. High rates of CRS have been reported, although they were manageable with the use of tocilizumab and corticosteroids. ICANS also occurs frequently, though typically at lower grades. Given the limited experience with CAR-T therapy in AML and the evolving landscape of these therapies, close monitoring of patients is essential. Additionally, priming strategies may help mitigate these toxicities by modulating the tumor burden or immune environment prior to CAR-T cell infusion.

Another concerning toxicity is myelotoxicity, due to the expression of many CAR-T target antigens on HSCs, which increases the risk of infectious and hemorrhagic complications. Gene-editing strategies, such as CD33 removal or CD45 gene knockout, could theoretically prevent this toxicity by eliminating the target antigen from HSCs, but these approaches have not yet been tested in human trials [106,107,108]. Alternatively, safety mechanisms like suicide genes and switchable CAR-T cells, are being developed to control CAR-T activity [109,110,111]. One such example is UniCAR-T-CD123, a switchable CAR-T tested in clinical trials is UniCAR-T-CD123 [69]. This product employs a modular system with a Targeting Module for CD123 (TM123), allowing for reversible CAR-T activation. Antitumor activity occurs only in the presence of both CAR-T cells and the TM, which can be rapidly cleared due to its short half-life. In this phase 1 trial, UniCAR-T-CD123 demonstrated a favorable safety profile, with all grade 3 CRS and grade 2 ICANS resolving within 24 h after interrupting TM administration. Importantly, no prolonged myelosuppression or need for subsequent allo-HSCT was observed.

Allogeneic CAR-T therapy poses the risk of resulting in GvHD or host allorejection [112,113]. To address these challenges, the AMELI-01 clinical trial investigated UCART123v1.2, an allogeneic anti-CD123 CAR-T product genetically modified to eliminate a functional T-cell receptor, thereby reducing the risk of GvHD [39]. Additionally, the CD52 gene was knocked out to enable the use of alemtuzumab as part of the lymphodepletion regimen, facilitating host immune suppression and minimizing allorejection of the CAR-T. The study reported that alemtuzumab was well tolerated and associated with enhanced CAR-T cell expansion and persistence.

### 4.6. Limited Efficacy Results and Role of Allo-HSCT

CAR-T therapy in R/R AML has demonstrated moderate preliminary efficacy, with significant variability in response rates across studies. Despite substantial trial heterogeneity and limited sample size, early studies suggest that targeting CD123, CLL-1, and CD7 could induce higher CR rates, while therapies against NKG2D-L have demonstrated modest efficacy. However, responses are often short-lived, and most patients eventually experience progression or relapse [39,62,72,74]. Moreover, long-term outcomes remain uncertain due to limited follow-up durations in most reports. It is important to note that these are mostly recent, early-phase studies, which underscores the need for more mature and longer-term data to fully evaluate the duration of response, long-term remission rates, and the potential late-onset toxicities. Given the transient nature of remissions and the potential for myeloablation, CAR-T therapy is often employed as a bridging strategy to allo-HSCT in many clinical trials [28,42,60,61,62,65,66,67,68,71,73,75,76,78,79,80]. For instance, in the study by Liu et al. [42], CAR-T therapy alone achieved only short-lived MRD-negative remissions (3.3–4.5 months), whereas patients who received subsequent allo-HSCT or local irradiation showed prolonged survival ranging from 10 to 26 months, suggesting that transplantation or other forms of consolidation following CAR-T therapy may be necessary to sustain long-term remissions. In R/R B-ALL, larger studies have evaluated allo-HSCT consolidation after CAR-T therapy, supporting the feasibility of this approach in heavily pretreated patients. For example, a retrospective study of 95 patients who relapsed post-transplant reported a 3-year OS of 55.3% and leukemia-free survival of 55.3%. The study also found that better outcomes were achieved when allo-HSCT was performed within 90 days after CAR-T [114]. However, these findings cannot be directly extrapolated to AML, given the biologically distinct nature of the disease. Additionally, this approach presents several challenges, including the need for timely donor availability and the risk of cumulative toxicities from CAR-T therapy itself, such as CRS and ICANS, and from allo-HSCT, including conditioning-related toxicity, GvHD, and infections. For instance, the anti-CLL1 CAR-T used by Zhao et al. [75] had a promising CR/CRi rate of 74.5%. However, prolonged grade 4 neutropenia required allo-HSCT in 28 patients (80%) to restore hematopoiesis. Another major limitation is that not all patients are eligible for allo-HSCT due to age or comorbidities, which increases the risk of relapse and prolonged cytopenias. Therefore, optimizing CAR-T therapies by identifying novel targets and minimizing toxicities is crucial to improve their applicability and achieve durable long-term outcomes.

## 5. Conclusions

CAR-T cell therapy shows promise as a treatment for R/R AML, with several clinical trials reporting encouraging results, especially those targeting CLL-1 and CD123. Research into new optimal target antigens is still ongoing. However, the advancement of CAR-T cell therapy for R/R AML remains still in its early stages, and it faces several major challenges. Efforts are being made to refine CAR-T cell design to enhance efficacy and safety, such as modifying the tumor microenvironment and developing dual and switchable CAR-T therapies. Additionally, novel strategies are being devised to address manufacturing challenges, such as the development of allogeneic CAR-T cells that are ready-to-use and widely available for clinical use. Nonetheless, further well-designed trials with larger sample sizes are needed to better assess toxicity profiles, determine the optimal lymphodepleting regimen, evaluate long-term response potential, and clarify the role of bridging to allo-HSCT. In particular, the use of CAR-T as a bridge to allo-HSCT in R/R AML, while promising and supported by preliminary clinical data, requires more evidence to define optimal tailored strategies and balance efficacy with cumulative toxicity. Ultimately, these advancements aim to expand curative treatment options for this difficult-to-treat patient population.

## Figures and Tables

**Figure 1 curroncol-32-00322-f001:**
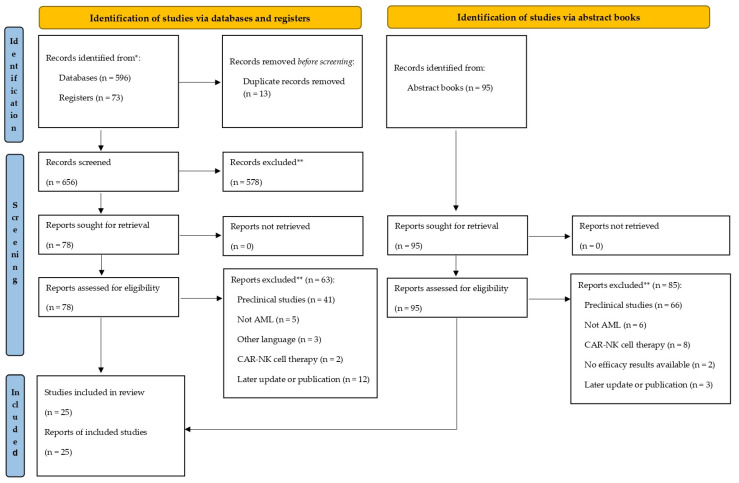
PRISMA 2020 flow diagram. * Databases: PubMed; registers: clinical trial registry (clinicaltrials.gov). ** Exclusions were applied following title and abstract screening. Records from registers (n = 73) were screened for results; none were added due to lack of available data or prior capture via PubMed.

**Figure 2 curroncol-32-00322-f002:**
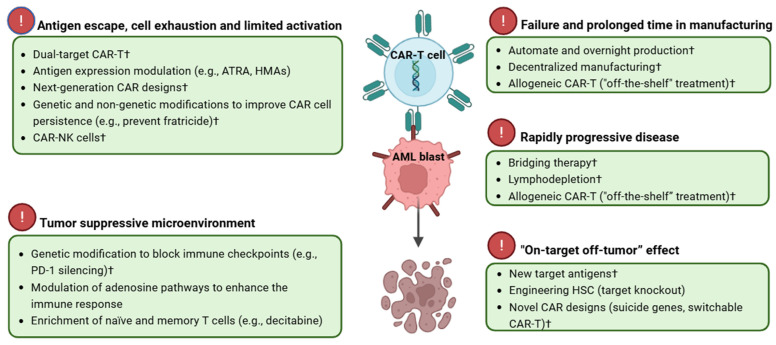
Summary of challenges and potential strategies for CAR-T therapy in acute myeloid leukemia. † Currently in clinical phase; all others are in preclinical phase. Image created with biorender.com. CAR-T: chimeric antigen receptor T cell; AML: acute myeloid leukemia; ATRA: all-trans retinoic acid; HMAs: hypomethylating agents; PD-1: programmed cell death protein 1; HSC: hematopoietic stem cell.

**Table 1 curroncol-32-00322-t001:** Description of main AML target antigens proposed for CAR T-cell therapy.

Target Antigen	Type of Molecule	Role	On HSC	On LSC	On AML Blasts	References	Clinical Trials ID
CD7	Glycoprotein	B and T-cell lymphoid interaction and development	No	Yes	20–35%	[24,27,28]	NCT05907603, NCT04033302, NCT04762485, NCT05995028, NCT04938115, NCT05513612, NCT04538599, NCT04599556
CD19	Glycoprotein	B-cell differentiation	No	No	8% ^a^	[42,43]	NCT04257175, NCT03896854, NCT05513612, NCT02772198
CD33	Glycoprotein (Siglec family)	Immune response, transmembrane receptor	Yes	Yes	90%	[23,24]	NCT06420063, NCT03222674, NCT05473221, NCT05467254, NCT05943314, NCT06326021, NCT05672147, NCT05016063, NCT01864902, NCT05945849, NCT03126864, NCT05248685, NCT04835519, NCT02799680, NCT05445765, NCT03971799, NCT05942599, NCT03291444, NCT03927261, NCT03795779
CD38	Glycoprotein	Cell adhesion, migration and signal transduction	No	Yes	58–91%	[32,39]	NCT03222674, NCT06110208, NCT03291444, NCT04351022
CD44v6	Variant 6 of the hyaluronic acid receptor CD44, glycoprotein	Leukocyte activation and malignant transformation	No	Yes	60–70%	[44,45]	NCT04097301
CD70	Member of TNF receptor superfamily	Immune response, transmembrane protein	No	Yes	90%	[29,30]	NCT04662294
CD84	Member of the SLAM family (SLAMF5)	Transmembrane Receptor	No	Yes	99%	[25,40]	NCT06786299
CD93	Glycoprotein	Immunological clearance of apoptotic cells, cell adhesion	No	Yes	55%	[34]	Preclinical studies
CD123	Type I IL-3 receptor alpha chain	Proliferation and survival of AML blasts	Yes	Yes	97%	[21,22,23]	NCT05995041, NCT03556982, NCT03796390, NCT03631576, NCT06420063, NCT04265963, NCT03222674, NCT06125652, NCT04318678, NCT04599543, NCT04106076, NCT02623582, NCT04272125, NCT04014881, NCT03672851, NCT04230265, NCT03190278, NCT02159495, NCT03291444, NCT05457010, NCT05513612
ADGRE2	Adhesion GPCR family	Cell adhesion	No	Yes	90–100%	[30,46]	NCT05463640
CLL-1	Glycoprotein	Inhibitory receptor, transmembrane receptor	No	Yes	80–90%	[24,47]	NCT03631576, NCT03222674, NCT05467254, NCT05943314, NCT05016063, NCT05248685, NCT06110208, NCT05467202, NCT04219163, NCT04884984, NCT04923919, NCT04789408, NCT06118788, NCT06017258, NCT03795779, ChiCTR2000041054
FLT3	Receptor tyrosine kinase III	Stem cell proliferation and differentiation	Yes	Yes	70–100%	[48,49]	NCT05023707, NCT03904069, NCT05266950, NCT05445011, NCT05432401, NCT05445011
FRβ	Folate-binding protein receptor	Folate delivery	No	Yes	70%	[33,50]	Preclinical studies, xenograft studies
IL1RAP	Glycoprotein	IL-1 signaling regulation	No	Yes	79%	[35,36,37,41]	NCT06281847, NCT04169022
ILT3 (LILRB4)	Leukocyte Ig-like Receptor-B Family	Inhibition of MHC class I immune activation	No	Yes	>90% ^b^	[51]	NCT04803929
Lewis Y	Glycoprotein	Embryogenesis, differentiation, tumor metastasis	No	Yes	50%	[52]	NCT01716364
NKG2D	C-type lectin-like receptor protein	Activating receptor of T and NK cells	No	Yes	75–80%	[53,54]	NCT04658004, NCT04167696, NCT03018405, NCT02203825
Siglec-6	SIGLEC Family (type I transmembrane glycoprotein)	Immune cell regulation and recognition	No	Yes	60%	[55,56]	NCT05488132
TIM-3	T-cell immunoglobulin mucin-3	Immune response	No	Yes	87%	[24,57]	NCT06125652
WT1	Zinc-finger transcription factor	Cell differentiation, proliferation and apoptosis	No	Yes	70–90%	[58,59]	NCT01640301

HSC: hematopoietic stem cells; LSC: leukemic stem cells; AML: acute myeloid leukemia; IL-3: interleukin 3; SIGLEC: sialic acid binding immunoglobulin-like lectin; TNF: tumor necrosis factor; SLAM: signaling lymphocytic activation molecule; FRβ: folate receptor β; FLT3: FMS-like tyrosine kinase 3; NKG2D: natural killer group 2 D; WT1: Wilms tumor 1; GPCR: G protein-coupled receptor; IL1RAP: interleukin-1 receptor accessory protein. ^a^ Frequently expressed as an aberrant marker in AML with t(8;21). ^b^ In monocytic/monoblastic AML (FAB M4/M5).

**Table 2 curroncol-32-00322-t002:** Characteristics of CAR-T constructs in published human trials in AML.

Target	Source	Country	Phase (ID)	Costim	Modification	Source	Transduction	Manufacturing	Ref.
CD7	Publication	China	I/II (NCT04762485)	–	–	Auto	–	–	[28]
	Publication	China	I (NCT04538599)	4-1BB	TCR, MHC, and CD7 KO +NKi + γc	Allo	Retroviral	–	[60]
	Abstract (EBMT 2023)	China	I (NCT04599556)	–	–	Auto	–	–	[61]
CD19	Publication	Israel	II (NCT02772198)	CD28	–	Auto	Retroviral	10 days	[62]
	Publication	China	–	4-1BB	–	Auto	Lentiviral	–	[42]
CD33	Publication	United States	I (NCT03126864)	4-1BB	–	Auto	Lentiviral	–	[63]
	Publication	China	I (NCT01864902)	4-1BB	–	Auto	Lentiviral	13 days	[64]
	Abstract (ASH 2022)	United States	I/Ib (NCT03927261)	–	Membrane-bound IL-15 + suicide switch	Auto	–	–	[65]
	Abstract (ASH 2023)	United States	I/II (NCT03971799)	–	–	Auto	–	–	[66]
	Publication	China	I/II (NCT04835519)	CD28	Potentiating molecule linked	Auto/Allo	Lentiviral	–	[67]
	Abstract (ASH 2024)	China	–	–	–	Auto/Allo	–	–	[68]
CD38	Publication	China	I/II (NCT04351022)	4-1BB	–	Auto/Allo	–	–	[39]
CD123	Abstract (ASH 2023)	Germany	I (NCT04230265)	CD28	Switchable uniCAR-T (TM123)	Auto	–	–	[69]
	Abstract (ASH 2022)	United States	I (NCT03190278)	–	TCR KO to reduce GvHD	Allo	Lentiviral	–	[70]
	Abstract (ASH 2017)	United States	I (NCT02159495)	CD28	–	Auto/Allo	Lentiviral	–	[71]
	Abstract (ASH 2022)	United States	I (NCT04318678)	CD28	–	Auto	Lentiviral	–	[72]
	Publication	China	–	4-1BB	–	Allo	Retroviral	8–12 days	[73]
CLL-1	Publication	China	I/II (NCT03222674)	4-1BB	–	Auto	Lentiviral	–	[74]
	Publication	China	I (ChiCTR2000041054)	4-1BB	–	Auto	Lentiviral	–	[75]
	Publication	China	I/II (NCT04884984)	CD28	PD-1 silenced	Allo	–	14 days	[76]
	Publication	China	–	CD28	–	Allo	Lentiviral	–	[77]
CLL-1-CD33	Abstract (EHA 2020)	China	I (NCT03795779)	–	Dual target	Auto/Allo	–	–	[78]
NKG2D-L	Publication	United States	I (NCT03018405)	DAP10	PI3K inhibition to prevent fratricide	Auto	–	–	[79]
	Publication	United States	I (NCT02203825)	DAP10	–	Auto	Retroviral	9 days	[80]
	Abstract (ASH 2020)	Belgium	I (NCT04167696)	DAP10	MICA and MICB KO to prevent fratricide	–	–	–	[81]

CoStim: co-stimulator domain; CAR-T: chimeric antigen receptor T cell; ASH: American Society of Hematology Congress; EHA: European Hematology Association Congress; EBMT: European Society for Blood and Marrow Transplantation Congress; PI3K: phosphoinositide 3-kinase; KO: knockout; NKG2D-L: natural killer cell group 2D ligand; CLL1: C-type lectin-like molecule-1; LeY Ag: Lewis Y antigen; MHC: major histocompatibility complex; MICA/MICB: MHC class I chain-related protein A/B; scFv: Single-Chain Variable Fragment; NKi: natural killer cell inhibitory receptor; γc: common cytokine receptor γ chain; TCR: T-cell receptor; Allo: allogeneic; Auto: autologous.

**Table 3 curroncol-32-00322-t003:** Disease and patient characteristics, efficacy and safety outcomes of CAR-T cell therapies in AML.

	Disease and Patient Characteristics					Conditioning and Dose		Safety Results		Efficacy Results		
Target	N	Median Age (Range)	EM (%)	Prior Lines (Range)	Prior HSCT (%)	LD	Dose (cells/kg)	CRS (%) [≥G3 (%)]	ICANS (%) [≥G3 (%)]	ORR (%) ^a^	Post-CART HSCT	Ref.
CD7	1	17	–	2	No	Cy + Flu + DEC	5 × 10^6^	Grade 3	0	MLFS	Yes	[28]
	12 ^b^	34 (8–66)	3/12 (25)	4 (2–7)	3/12 (25)	Cy + Flu + Eto	(1–3) × 10^7^	10/12 (83) [0]	0	1/1 (100)	4 pts	[60]
	9	–	–	–	–	Cy + Flu/Eto	(2–4) × 10^6^	9/9 (100) [0]	1/9 (11) [0]	6/9 (66.7)	3 pts	[61]
CD19	6 ^c^	–	1/6 (17)	4 (3–8)	4/6 (66)	Cy + Flu	1 × 10^6^	6/6 (100) [1(16.6)]	2/6 (33) [0]	4/6 (66.7)	2 pts	[62]
	8 ^d^	28 (5–40)	–	–	4/8 (50)	Cy + Flu	Median 1.5 × 10^5^	7/8 (87) [1 (12.5)]	0	5/8 (62.5)	4 pts	[42]
CD33	10	30 (18–73)	–	5 (3–8)	3/10 (30)	None	0.3 × 10^6^	2/3 (66.7) [1 (33.3)]	1/3 (33) [0]	0	No	[63]
	1	41	–	–	No	None	1.12 × 10^9^	Grade 4	0	0	No	[64]
	24	60 (33–77)	–	3 (1–9)	15/24 (50)	None/Cy + Flu	3 × 10^4^–10^6^	17/20 (85) [1 (5)]	–	2/20 (10)	1 pt	[65]
	24	16 (1–34)	–	4 (2–6)	3/4 (75)	Cy + Flu	3 × 10^5^–1 × 10^7^	13/19 (68) [4 (21)]	1/19 (5.3) [1 (5.3)]	2/19 (11)	1 pt on day 100	[66]
	4	9.5 (3–12)	No	–	3/4 (75)	Cy + Flu	5×10^5^ (±20%)	4/4 (100) [1 (25)]	2/4 (50) [0]	3/4 (75) ^e^	3 pts on day 29–38	[67]
	12	36 (14–51)	4/12 (33)	–	12/12 (100)	Flu regimen	6.2×10^4^–6.15 × 10⁵	10/12 (83) [0]	0	5/12 (41.7)	5 pts	[68]
CD38	6	34.5 (7–52)	–	–	6/6 (100)	Cy + Flu	(6.1–10) × 10^6^	4/6 (66.7) [1 (16.6)]	0	4/6 (66.7)	No	[39]
CD123	19	–	–	4 (2–7)	12/19 (63)	Cy + Flu	<20 × 10^7^	12/19 (63) [3 (15.7)]	1/19 (5) [0]	8/15 (53)	No	[69]
	16	57 (18–65)	–	4 (3–9)	9/16 (56)	Cy + Flu ± Alem	(0.25–3.03) × 10^6^	15/16 (95) [3 (18.7)]	1/16 (6.25) [1 (6.25)]	2/16 (12.5)	–	[70]
	14	–	1/7 (14)	4 (4–7)	6/7 (86)	Cy + Flu	DL1:5 × 10^7^; DL2: 2 × 10^8^	5/7 (71.4) [0]	0	3/6 (50)	2 pts	[71]
	12	17 (12–21)	1/12 (8)	–	11/12 (92)	Cy + Flu	(3–100) × 10^5^	– [0]	– [0]	2/5 (40)	–	[72]
	1	25	–	9	Yes	RIC regimen of TVFB	1.1 × 10^8^	Grade 3	–	CRi	Yes, at day 6	[73]
CLL-1	8	12 (8–16)	–	3 (1–6)	2/8 (25)	Cy + Flu	(0.35–1) × 10^6^	8/8 (100) [0]	0	6/8 (75)	6 pts	[74]
	47	35 (17–73)	20/47 (43)	4 (2–13)	14/47 (30)	Cy + Flu	(0.5–3) × 10^6^	45/47 (96) [25 (53)]	10/47 (21.3) [3 (6.4)]	35/47 (74.5)	30 pts	[75]
	2	28 (both)	–	Pt 1: 2; Pt 2: 4 ^f^	2/2 (100)	Cy regimen	Pt 1: 1 × 10^7^; Pt 2: 5 × 10^6^	Pt 1: G1, Pt 2: G2	0	2/2 (100)	1 pt	[76]
	1	18	No	3	Yes	Cy + Flu	0.5 × 10^6.^	Grade 1	0	CR	Yes	[77]
CLL-1-CD33	9	32 (6–48)	–	–	–	Cy + Flu	(1–3) × 10^6^	8/9 (88.8) [2 (22.2)]	4/9 (44) [3 (33.3)]	7/9 (78)	6 pts	[78]
NKG2D	25	67 (57–76)	–	1 (1–2)	–	None	(3–30) × 10^8^	15/16 (94) [5 (31.2)]	0	3/12 (25)	2 pts	[79]
	14	70 (44–79)	–	1 (0–4)	No	None	1 × 10^6^–3 × 10^7^	None	0	0	1 pt	[80]
	11	–	–	–	–	Cy + Flu	(1–10) × 10^9^	– [5 (55.5)]	–	0	–	[81]

CAR-T: chimeric antigen receptor T cell; AML: acute myeloid leukemia; EM: extramedullary disease; LD: lymphodepletion; CRS: cytokine release syndrome; ICANS: immune effector cell-associated neurotoxicity syndrome; ORR: overall response rate; HSCT: hematopoietic stem cell transplantation; NKG2D-L: natural killer cell group 2D ligand; CLL1: C-type lectin-like molecule-1; LeY Ag: Lewis Y antigen; R/R: refractory/relapsed; Pts: patients; Cy: cyclophosphamide; Flu: fludarabine; RIC: reduced-intensity regimen; TVFB: therarbucin, teniposide, fludarabine and busulfan; Eto: etoposide; Alem: alemtuzumab; DL: dose level. ^a^ Among evaluable patients. ORR encompasses complete remission (CR), complete remission with incomplete hematologic recovery (CRi), and morphologic leukemia free state (MLFS). ^b^ Only one patient was diagnosed with AML. ^c^ Five out of six AML patients had t(8;21). ^d^ Four out of seven patients with AML had t(8;21). ^e^ One patient who initially did not respond received a second infusion and achieved CRi with MRD-positive. ^f^ Both including prior anti-CD33 CART therapy.

## Data Availability

The data presented in this study are available in this article and Appendix A.

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
