# Peer review of "CAR-T Cell Therapy for Acute Myeloid Leukemia: Where Do We Stand Now?"

_curroncol, 2025, doi:10.3390/curroncol32060322_

Round 1

Reviewer 1 Report

Comments and Suggestions for Authors

In this manuscript, the authors review the clinical applications of both autologous and allogeneic CAR T-cell therapies in leukemia. The manuscript is well-written, and the authors have provided a thorough and detailed methodology for their literature search. Below are a few minor suggestions for improvement:

  1. The authors are encouraged to elaborate further on the design strategies of allogeneic CAR T cells, highlighting their potential advantages over autologous CAR T cells, as well as the associated challenges.

  2. While the primary focus of the article is on CAR T cells, the authors may consider briefly discussing recent advances in CAR NK cell therapies to provide a comparative perspective.

Reviewer 2 Report

Comments and Suggestions for Authors

Summary

This systematic review offers a thorough examination of the current clinical picture for CAR-T cell therapy in relapsed/refractory acute myeloid leukemia. It presents findings from 25 clinical investigations and trials involving 296 patients, with an emphasis on target antigens, CAR-T designs, response rates, toxicities, and the role of allogeneic hematopoietic stem cell transplantation (allo-HSCT). The review shines at describing clinical outcomes, identifying important difficulties, and recommending future strategies for increasing CAR-T efficacy and safety in AML.

General Concept Comments

The review tackles a pressing clinical need through a highly relevant and timely issue. The authors effectively combine clinical evidence from several trials, which is commendable given the fragmented and early-stage nature of most investigations. The organization is logical, and the flow is consistent. However, the study might benefit from a more in-depth examination of knowledge gaps, a more critical assessment of methodology across trials, and a clearer discussion of the translational implications of ongoing breakthroughs (for example, switchable CARs and allogeneic platforms).

Article-Specific Scientific Content Comments

  1. Hypothesis/Testability: While a hypothesis is not required for a review, the authors infer that CAR-T therapy can be used in R/R AML and serve as a bridge to allo-HSCT. The combined clinical data supports this position. A more clearly expressed central thesis, as well as a targeted review of whether current evidence supports this translational paradigm, would increase coherence.
  2. Methodological Clarity: The PRISMA procedure and inclusion criteria are clearly defined. However, there is very little consideration of study quality or potential biases. Given the reliance on early-phase trials and conference abstracts, a systematic assessment of bias risk (e.g., by trial phase, sample size, or reporting completeness) would improve rigor.
  3. Missing Controls and Limitations: The article acknowledges several limitations, such as the lack of RCTs and reliance on short-term follow-up. However, more attention should be focused on the lack of control arms in most studies and how this constrains interpretation of efficacy, especially when compared to historical salvage therapy.

Review-Specific Comments

  1. Completeness and Relevance: The review is comprehensive and up to date. It encompasses a wide range of targets (CD33, CD123, CLL-1, etc.) and constructs (such as switchable CARs, PD-1 knockouts, and dual targeting). It also addresses critical translational issues (toxicity, production, and antigen escape). However, it lacks a more in-depth explanation of resistance mechanisms, standardization concerns in response definition (e.g., MRD vs. CRi), and limits with long-term follow-up.
  2. Knowledge Gaps: While the article mentions challenges like myelotoxicity and manufacturing delays, it could better define open research questions. For example:
    • Are there specific immune or genetic features that predict CAR-T failure in AML?
    • What are the optimal antigen combinations to reduce relapse and avoid HSC toxicity?
    • Can CAR-T monotherapy replace bridge-to-transplant strategies in certain AML subsets?
  3. References: Most referenced research are current and relevant, with little self-citation. However, incorporating high-impact reviews or meta-analyses of CAR-T in other myeloid malignancies could provide a comparative perspective. More references to functional studies (e.g., CAR T cell exhaustion in AML models) would help to enrich mechanistic debates.

Specific Comments (Line/Table/Figure Reference)

  1. Lines 106–108 (Outcome definitions): Clarify if ORR includes both MRD-negative and MRD-positive responses or just morphological criteria.
  2. Figure 2: Excellent schematic summarizing challenges and strategies. Consider explicitly labeling which strategies are in clinical use vs. preclinical.
  3. Table 3: Very informative, but hard to digest due to density. Consider separating toxicity from efficacy for clarity. Also, define “EM” in the table legend.
  4. Line 233: In discussing myelotoxicity, state more clearly how the presence of allo-HSCT may obscure assessment of hematopoietic recovery.
  5. Lines 421–429: The role of allo-HSCT is well stated, but it would help to cite data on allo-HSCT in CAR-T-treated ALL as a benchmark for AML.
